# Synthesis of Transition Metal Complexes and Their Effects on Combustion Properties of Semi-Rigid Polyvinyl Chloride

**DOI:** 10.3390/ma14102634

**Published:** 2021-05-18

**Authors:** Pei Xiang, Jun Xu, Biao Li, Weiqi Liu, Jinshun Zhao, Qining Ke, Siwen Bi, Xuhuang Chen

**Affiliations:** 1School of Materials and Chemical Engineering, Hubei University of Technology, Wuhan 430068, China; 13407196501@163.com (P.X.); xujun950915@163.com (J.X.); li105695@163.com (B.L.); 18347947961@163.com (W.L.); qq2816964803@sina.com (J.Z.); ktycoon@163.com (Q.K.); siwen_bi@student.uml.edu (S.B.); 2Hubei Provincial Key Laboratory of Green Materials for Light Industry, Hubei University of Technology, Wuhan 430068, China

**Keywords:** polyvinyl chloride, thermal stability, flame retardancy, smoke suppression, mechanical properties

## Abstract

Using introduction of MoO_4_^2−^ and Fe^3+^, Cu^2+^, or Zn^2+^ into amphiphilic polymers (DN) via an ion-exchange reaction, different transition metal complexes, as retardants and smoke suppressants, including (DN)Mo, Fe(DN)Mo, Cu(DN)Mo, and Zn(DN)Mo were synthesized. Combined with the results of X-ray photoelectron spectroscopy (XPS), X-ray diffraction (XRD), and Fourier transform infrared spectroscopy (FTIR), it could be determined that ionic bonding of these ions with DN occurred. Subsequently, the influence of flame-retardant, smoke-suppression, and mechanical properties of (DN)Mo, Fe(DN)Mo, Cu(DN)Mo, and Zn(DN)Mo on polyvinyl Chloride (PVC) were tested. It was demonstrated that transition metal complexes of three metal elements, Fe(DN)Mo, Cu(DN)Mo, and Zn(DN)Mo, showed better flame retardancy, smoke suppression, and thermal stability as confirmed by microcalorimetry, limiting oxygen index (LOI), smoke density, and thermogravimetric analysis (TGA) tests, in which Cu(DN)Mo worked best due to the Lewis acid mechanism and reductive coupling mechanism. Scanning electron microscopy (SEM) showed that the addition of (DN)Mo, Fe(DN)Mo, Cu(DN)Mo, and Zn(DN)Mo promoted the formation of a dense carbon layer on the PVC surface during combustion, which could protect the interior PVC. The addition of these transition metal complexes hardly impaired the mechanical properties of PVC.

## 1. Introduction

Polyvinyl chloride (PVC) is one of the most commonly used thermoplastics and is widely applied in the fields of furniture, building materials, textiles, and transportation due to excellent mechanical properties, good chemical resistance, superior electrical insulation, and low price [1,2,3,4]. Although pure PVC has excellent flame retardancy, its smoke-suppression property is rather weak. The inevitable use of plasticizers reduces the flame retardancy of PVC and further profoundly deteriorates the smoke-suppression ability [5]. Therefore, it is urgent to develop efficient flame retardants and smoke suppressants for PVC.

Applications and mechanisms of PVC flame retardancy and smoke suppression have been extensively studied [6,7]. In terms of flame retardancy, common flame retardants are phosphorus-based compounds (i.e., red phosphorus and melamine phosphate), silicon-based compounds (i.e., silicone and polysilsesquioxane), and nitrogen-based compounds (i.e., melamine cyanurate) [8,9,10]. In particular, nitrogen-based flame retardants tend to produce NH_3_, N_2_, N_x_O_y_, and H_2_O during combustion, which can reduce the concentration of oxygen and the surface temperature of polymers, and exhibit superior flame retardancy [11]. With respect to smoke suppression, molybdenum compounds (i.e., molybdenum trioxide and ammonium molybdate) have garnered intensive attention [12]. It was reported that metal elements such as iron, zinc, and copper possess the ability of smoke suppression, and these metal elements have a mutual promotion effect with molybdenum in smoke suppression [13].

Nowadays, inorganic metal compounds are the most used flame retardants and smoke suppressants. Although these flame retardants and smoke suppressants possess smoke-suppression effect, stability, and low toxicity or nontoxicity, they usually have poor compatibility with a polymer matrix, leading to uneven particle size distribution, which impairs the flame retardancy, smoke suppression, and the mechanical properties of the matrix. In contrast, organic flame retardants and smoke suppressants usually have a natural advantage of compatibility with a polymer matrix. Meanwhile, their adverse effect on the mechanical performance of the matrix is relatively minor [6,14]. However, these additives show more or less weak smoke suppression, due to the lack of metal elements. For example, Jia et al synthesized a chlorinated phosphate based on soybean oil (CPSO) [15]. Although CPSO had a good dispersity in a PVC matrix, its smoke-suppression effect was indeed rather weak.

Recently, it was found that the incorporation of smoke-suppressing metal elements in organic flame retardants can address the above dilemma. Chen and coworkers synthesized four M–Phyt salts (M represents Cu, Zn, Al, or Sn) by direct precipitation [16]. The test results showed that the limiting oxygen index (LOI) of PVC/Sn–Phyt increased from 24.9% to 30.3%, while it had the lowest loss of mechanical properties. However, there have been few studies on the mutual promotion effect of two or more metal elements in organic flame retardants and smoke suppressants.

In this work, we reveal a novel strategy in which MoO_4_^2−^, MoO_4_^2−^/Fe^3+^, MoO_4_^2−^/Cu^2+^, or MoO_4_^2−^/Zn^2+^ were incorporated into an amphoteric polymer (DN) through an ion-exchange reaction. To evaluate the bonding of metal ions to the amphoteric polymer, the synthesized products were characterized by X-ray photoelectron spectroscopy (XPS), X-ray diffraction (XRD), and Fourier transform infrared spectroscopy (FTIR). Subsequently, the effect of (DN)Mo, Fe(DN)Mo, Cu(DN)Mo, and Zn(DN)Mo on flame retardancy, smoke suppression, compatibility, and mechanical properties of PVC were then further tested. This experiment proposes a new and feasible route for the synergistic enhancement of flame retardancy and smoke suppression effect by multimetal ions.

## 2. Experimental

### 2.1. Materials

PVC (SG-5, TL-1000) was used as received from Inner Mongolia Yili Chemical Industry Co., Ltd. (Dalad Banner, China). Organotin thermal stabilizer was obtained from Hubei Benxing Chemical Co., Ltd. (Suizhou, China). Calcium stearate was supplied by Chengdu Shengteng Technology Development Co., Ltd. (Chengdu, China). Dioctyl phthalate (DOP) was obtained from Wuhan Delip Chemical Co., Ltd. (Wuhan, China). Sodium 4-vinylbenzenesulfonate (NaSS), [2-(acryloyloxy)ethyl]trimethylammonium chloride solution (DMAEA-Q), N, N’-methylenebis(acrylamide) (MBAA), α-Ketoglutaric acid (KA), sodium molybdate dihydrate (Na_2_MoO_4_), ferric chloride (FeCl_3_), zinc chloride (ZnCl_2_), and copper chloride (CuCl_2_) were purchased from Shanghai McLean Biochemical Technology Co., Ltd. (Shanghai, China).

### 2.2. Synthesis of Transition Metal Ion Complexes

#### 2.2.1. Synthesis of an Amphoteric Polymer (DN)

A 60 mL beaker equipped with magnetic stirrer bar charged with DMAEA-Q (2.9 g), NaSS (2.6 g), MBAA (0.004 g), KA (0.004 g), and deionized water (total volume fixed to 10 mL). After reaching thermal equilibrium in a water bath at 60 °C, the mixture was stirred until it was completely dissolved. After that, the mixed solution was poured to the 2 mm-thick glass mold. Subsequently, the glass mold was placed under an ultraviolet lamp for 10 h to achieve photocatalytic copolymerization. The product was recorded as DN, which can react with metal ions in ion exchange [17].

#### 2.2.2. Synthesis of Molybdate Ion Complex ((DN)Mo)

The DN was soaked in Na_2_MoO_4_ solution (1 mol/L, 50 mL) for 3 days, ensuring that the ion exchange for MoO_4_^2^^−^ to bond with C_4_H_11_N^+^ was in balance. The synthetic route is shown in Figure 1; the product was named (DN)Mo.

#### 2.2.3. Synthesis of Transition Metal Complexes (A(DN)Mo)

The (DN)Mo was soaked in A^+^ Cl^−^ solution (1 mol/L, A^+^=Fe^3+^, Cu^2+^, or Zn^2+^), ensuring that the ion exchange for A^+^ to bond with C_6_H_5_O_3_S^−^ was in balance. The synthetic route is shown in Figure 1; the three products were recorded as Fe(DN)Mo, Cu(DN)Mo, and Zn(DN)Mo, respectively.

### 2.3. Preparation of PVC Composites

The basic formula of the PVC composites was 100 parts per hundred of resin (phr) PVC, 1 phr organic tin stabilizer, 20 phr DOP, and 0.4 phr calcium stearate. The addition of (DN)Mo, Fe(DN)Mo, Cu(DN)Mo, or Zn(DN)Mo was 8 phr. These components were melt-mixed with a two-roll mill (XH-401AE, Dongguan, China) at 160 °C for 5 min. After that, the samples were compressed into a sheet at 160 °C and 10 MPa for 8 min, and cooled to room temperature. The sheet was cut into the desired shapes and sizes for further testing.

### 2.4. Characterization

#### 2.4.1. X-ray Photoelectron Spectroscopy Test

The synthesized samples were characterized by X-ray photoelectron spectroscopy (XPS, PHI5000, JEOL, Beijing, China), using Al Kα (1486.6 eV) as the X-ray source with a high voltage of 12 kV, an emission current of 20 mA, and a fluence energy of 20 eV. Before processing the XPS spectra, the standard peak spectra of C1s were used to correct.

#### 2.4.2. X-ray Diffraction Test

The synthesized samples were analyzed by X-ray diffraction (XRD, Empyrean, Malvern Panalytical, Shanghai, China) with Cu target, Kα radiation, *λ* = 0.1542 nm, scan angle of 10° to 25°, scan rate of 1°/min, voltage of 40 kV, and current of 40 mA.

#### 2.4.3. Fourier Transform Infrared Spectroscopy Test

The synthesized samples were examined with a spectrometer (FTIR, Vertex 70, Bruker, Karlsruhe, Germany) using KBr pellets.

#### 2.4.4. Thermogravimetric Analyses

Thermogravimetric analyses (TGA) of the synthesized samples and PVC composite samples were performed using a TGA Q5000 thermal gravimetric analyzer (TA instruments, New Castle, DE, USA). The weight of all samples was about 5 mg. Each sample was tested in an Al_2_O_3_ pan, and the experimental temperature range was from 20 °C to 750 °C at a heating rate of 20 °C /min in a nitrogen atmosphere.

#### 2.4.5. Scanning Electron Microscopy

The morphologies of the synthesized samples, burned carbon layers, and PVC composites were all characterized by field-emission scanning electron microscopy (SEM, SU8010, Hitachi, Chiyoda City, Japan) with an accelerating voltage of 5 kV. The synthesized sample particles were distributed on conductive adhesive tape for SEM observation. The surfaces and cross-sections were obtained in the burned carbon layers which were derived from the samples after a limiting oxygen index test. PVC composites were freeze-fractured in liquid nitrogen. All surfaces to be tested were sputtered-coated with a layer of gold (deposition time 20 s) before observation.

#### 2.4.6. Smoke Density Test

The smoke densities (Ds) of the PVC composites were measured in a smoke density chamber (FTT 0064, Earth Products China Limited, Hong Kong, China) following the procedure of the ISO 5959-2: 2006 standard with an external heat flux of 25 KW/m^2^ and flameless combustion. The dimensions of the tested specimens were 75 mm × 75 mm × 1 mm. According to the standard, Ds_600_ is the Ds value at 600s and Ds_max_ is the maximum Ds value during the 600 s of combustion.

#### 2.4.7. Limiting Oxygen Index

The limiting oxygen index (LOI) of the PVC composites was measured by oxygen index meter (HC-2, Jiangning Testing Equipment, Nanjing, China) in sheet dimensions of 120 mm × 10 mm × 4 mm following GB/T 2406.2-2009 standards.

#### 2.4.8. Microcalorimetry Test

The actual combustion of the PVC composites was measured by microcalorimetry (MCC, FTT 0001, Earth Products China Limited, Hong Kong, China), with samples weighing about 5 mg. The heating temperature was raised to 750 °C at a heating rate of 2 °C/s in a stream of oxygen flowing at 20 mL/min. The combustion temperature was set at 900 °C, and the oxygen/ nitrogen flow rate was set at 20 mL/80 mL.

#### 2.4.9. Mechanical Properties

Impact strength of samples in the dimensions of 80 mm × 10 mm × 4 mm was tested according to the GB/T1043.1-2008 standard using an electronic impact tester (GT-7045-MDL, GTS systems co., Ltd., Dongguan, China). Tensile strength of dumbbell-shaped PVC composites samples in the dimensions of 75 mm × 4 mm × 2 mm was tested using an electronic tensile tester (GMT6130, MTS systems Co., Ltd., Eden Prairie, MN, USA) according to the procedure of the GB/T1040.2-2006 standard. Tests were conducted at an extension rate of 5.0 mm/min at 25 °C. Flexural strength of PVC composites samples in the dimensions of 80 mm × 10 mm × 4 mm was tested using an electronic tensile tester (GMT6130, MTS systems Co., Ltd., Eden Prairie, MN, USA) according to the procedure of the GB/T 9341-2008 standard. For each result, an average value from a minimum of five test specimens were report together with standard deviations.

## 3. Results and Discussion

### 3.1. Characterization of Flame Retardants and Smoke Suppressants

Figure 2 shows the XPS spectra of (DN)Mo, Fe(DN)Mo, Cu(DN)Mo, and Zn(DN)Mo. As shown, XPS signals at 164 eV, 285 eV, 402 eV, and 530.6 eV corresponded to S 2p, C 1s, N 1s, and O 1s of the organic matrix in flame retardants and smoke suppressants, respectively, as reported in previous works [18,19]. The XPS signals at 232.1 eV (corresponding to Mo 3d), 706.7 eV (corresponding to Fe 2p3), 932.5 eV (corresponding to Cu 2p3), and 1021.8 eV (corresponding to Zn 2p3) indicated that Mo, Fe, Cu, and Zn were introduced into the DN.

Figure 3 shows that the XRD diffraction patterns of (DN)Mo, Fe(DN)Mo, Cu(DN)Mo, and Zn(DN)Mo had weak signals, indicating that the synthesized samples were amorphous and showed a low and poorly ordered crystallinity [20]. Combined with the XPS results, it could be concluded that the ionic bonding took place between MoO_4_^2−^, Fe^3+^, Cu^2+^, or Zn^2+^ and DN, rather than DN simply physically adsorbing inorganic metal salts such as FeCl_3_, CuCl_3_, and ZnCl_2_, proving the occurrence of the desired ion-exchange reactions.

The FTIR spectra of DN, (DN)Mo, Fe(DN)Mo, Cu(DN)Mo, and Zn(DN)Mo are shown in Figure 4. It can be seen that the absorption peaks between 1400 cm^−1^ and 3500 cm^−1^ were basically similar when comparing DN with the four transition metal ion complexes. From the IR spectrum of DN, it was found that the asymmetric absorption peak of the main chain methylene C-H appeared near 2926 cm^−1^, and the stretched vibration peak of the quaternary ammonium cation unit ester bond C=O appeared at 1731 cm^−1^, which indicated that DN had been produced by a polymerization reaction. Broad absorption peaks at 3444 cm^−1^ and 1636 cm^−1^, corresponding to -OH bonds, probably were due to crystalline water or adsorbed water in the DN [21]. Then, the change of functional group before and after the ion-exchange reaction were analyzed. The C-N stretching vibration peak appeared near 1400 cm^−1^, and the C-N stretching vibration peak of four transition metal complexes shifted toward high wavenumbers after the reaction with MoO_4_^2−^. In addition, the four transition metal complexes had a strong absorption peak at 840 cm^−1^, 839 cm^−1^, 840 cm^−1^, and 838 cm^−1^, respectively, while the absorption peak of the Mo=O bond was located around 964 cm^−1^. This indicated that the quaternary ammonium group had interacted with MoO_4_^2−^, which proved the success of the ion-exchange reaction. The stretched vibrational peak of the -S=O group in the sulfonic acid group was located at 1186 cm^−1^. After the reaction with Fe^3+^, Cu^2+^, and Zn^2+^, the stretched vibrational peak of the -S=O group for Fe(DN)Mo, Cu(DN)Mo, and Zn(DN)Mo shifted toward low wavenumbers, which indicated that the metal ions interacted with the -S=O group of the sulfonic acid group [22].

The mass losses of (DN)Mo, Fe(DN)Mo, Cu(DN)Mo, and Zn(DN)Mo in TGA are shown in Figure 5. It can be seen that the thermal degradation process consisted of two stages. The first decomposition stage occurred from 30 °C to 280 °C, and the weight loss was about 10%, which was attributed to the evaporation of water vapor [23,24]. The second stage proceeded from 300 °C to 480 °C, which corresponded to the degradation of the organic matter matrix and the production of volatile gases such as CO, CO_2_, and NH_3_. The residual was mainly composed of transition metal compounds and carbon. It was noteworthy that (DN)Mo, Fe(DN)Mo, Cu(DN)Mo, and Zn(DN)Mo showed residual content at 700 °C of up to about 53.1 wt %, 59.3 wt %, 64.2 wt %, and 64.1 wt %, respectively, which indicated these transition metal complexes had sufficient thermal stability to meet the processing temperature of plastics.

Figure 6 shows the SEM micrographs of (DN)Mo, Fe(DN)Mo, Cu(DN)Mo, and Zn(DN)Mo. As shown, the four synthesized samples showed no obvious agglomeration. This attribute is critical to maintaining the mechanical properties of PVC, which will be discussed later.

### 3.2. Characterization of Smoke Density for PVC Composites

The (Ds) curves of PVC, PVC/(DN)Mo, PVC/Fe(DN)Mo, PVC/Cu(DN)Mo, and PVC/Zn(DN)Mo are shown in Figure 7, and the corresponding Ds_max_ and Ds_600_ are summarized in Table 1. Compared with pure PVC, the addition of (DN)Mo significantly decreased the Ds_max_ and Ds_600_ of PVC, which indicated that (DN)Mo could improve the smoke-suppression performance of the PVC. Mo could react with the HCl decomposed from PVC during combustion, which tends to form cross-linking of polyene chains catalyzed by strong Lewis acid [6]. This behavior can promote charring of PVC, and reduce smoke emission. Notably, compared with PVC/(DN)Mo, PVC/Fe(DN)Mo, PVC/Cu(DN)Mo, and PVC/Zn(DN)Mo showed lower Ds_max_ and Ds_600_, indicating that two metal elements of Fe-Mo, Cu-Mo or Zn-Mo exhibited more outstanding smoke-suppression performance. This was due to the fact that Fe and Zn have a mutual promotion effect with Mo by the Lewis acid mechanism [25]. PVC/Cu(DN)Mo also showed the optimum effect of smoke suppression due to the fact that Cu works in smoke suppression mainly through a reductive coupling mechanism, and by a synergistic contribution of the Lewis acid mechanism [12,26]. In conclusion, all transition metal ion complexes had excellent smoke-suppression efficiency.

### 3.3. Characterization of Flame Retardancy for PVC Composites

LOI is an important measurement of the flame-retardant properties of a material as a reference value. The combustion behavior of the composites is often characterized by MCC, which is widely used to evaluate the flammability of polymer materials [27]. Several important combustion parameters, including heat-release rate (HRR), peak value of heat release (pHRR), time at pHRR (T_p_), and total heat release (THR) can be obtained from MCC. The curves of HRR and THR versus time for pure PVC and the PVC composites are shown in Figure 8.

As shown in Figure 8, the HRR curves had two peaks, which was consistent with the reported results [28]. The first peak was mainly related to HCl emission and DOP degradation, while the second peak was the main chain cracking of PVC and production of some low molecular weight gaseous compounds [29]. It was reported that the HCl removal reaction, crosslinking of PVC, and carbon production can by catalyzed by a metal element based on a strong Lewis acid mechanism [30].

The combustion parameters are summarized in Table 2. The two stages of the pHRR and THR of the PVC composites was much lower than those of pure PVC, as shown in Table 2. This is because the formation of stable carbon on the PVC surface during combustion could effectively protect the interior from burning and heat, which could reduce the heat released throughout the combustion process. Notably, the pHRR in the second peak and THR of PVC/Cu(DN)Mo had the lowest values of 36.45% and 48.89%, respectively, compared with that of PVC/(DN)Mo, PVC/Fe(DN)Mo, and PVC/Zn(DN)Mo. This was because Cu and Mo have a strong synergistic effect in smoke suppression [12]. Compared with PVC/(DN)Mo, the LOI of the PVC/Fe(DN)Mo, PVC/Cu(DN)Mo, and PVC/Zn(DN)Mo materials were all higher, showing that the use of Fe, Cu, or Zn in combination with Mo had superior flame-retardant efficiency.

### 3.4. Thermal Stability of PVC Composites

The mass losses of PVC, PVC/(DN)Mo, PVC/Fe(DN)Mo, PVC/Cu(DN)Mo, and PVC/Zn(DN)Mo are presented in Figure 9, and the corresponding thermogravimetric parameters are illustrated in Table 3. The thermal degradation of the samples showed two stages, as shown in Figure 9. In the first stage, the side chains of PVC decomposed to produce HCl. In the second stage, the PVC main chain was cleaved to produce low molecular weight gases [31].

As presented in Table 3, the temperature at 5% mass loss (T_5%_) of PVC was 273.71 °C. In the first stage, the mass loss was 66.05%, and the corresponding temperature of maximum mass loss rate (T_max_) was 319.02 °C. In the second stage, the mass loss was 27.12% and the T_max_ was 468.66 °C, with a final char residue of 5.94%. Compared with PVC, the T_5%_ and mass loss of PVC/(DN)Mo in the second stage decreased, and the final solid residue increased significantly. This indicated that (DN)Mo had a superior charcoal-forming ability, which facilitated the formation of stable carbon slag and improvement of the thermal stability for PVC at a high temperature. It can be seen that the T_5%_ and mass losses of the PVC/Cu(DN)Mo, PVC/Fe(DN)Mo, and PVC/Zn(DN)Mo composites were reduced compared with that of PVC/(DN)Mo. This indicated that the PVC/Fe(DN)Mo, PVC/Cu(DN)Mo, and PVC/Zn(DN)Mo composites had good thermal-stability performance. Since Fe^3+^ and Zn^2+^ could react with HCl to produce FeCl_3_ and ZnCl_2_, FeCl_3_ and ZnCl_2_ as a Lewis acid in condensed matter could catalyze the removal of HCl from PVC and promote earlier cross-linking of PVC, leading to rapid carbonization. The PVC/Cu(DN)Mo composite had the best thermal-stability performance. This was because the Cu in PVC/Cu(DN)Mo catalyzed the elimination of allyl chloride groups through a reductive coupling reaction, which inhibited the removal HCl reaction of “zipper-type” chain and slow down the thermal decomposition of PVC. Nitrogen-based flame retardants tend to produce NH_3_, N_2_, N_x_O_y_, and H_2_O during combustion, which can reduce the concentration of oxygen and the surface temperature of a polymer, contributing to flame retardancy [11].

### 3.5. Morphology Analysis of Char Residue

The morphology of surfaces and cross-sections for the char residues after the LOI test is shown in Figure 10. As shown in Figure 10(a1), many small holes existed on the surface of the PVC, which may have resulted in less protection for the matrix during combustion. This was because the dense carbon layer had the ability to insulate from heat and flames, while the presence of holes made the carbon layer less protective. As shown in Figure 10(a2,a3), numerous large holes existed in the cross-section, which confirmed the inadequacy of the porous carbon layer on the surface to protect the internal matrix. The surface of PVC/(DN)Mo was relatively flat and dense without holes, as shown in Figure 10(b1). This was due to the fact that the addition of (DN)Mo promoted the cross-linking of the PVC and the formation of a dense carbon layer, which was in agreement with the results of a higher char residue in TGA. The flat and dense characteristics were beneficial to isolating the contact between the external flame or heat and the internal matrix, which could be confirmed by the smaller and fewer holes shown in Figure 10(b2,b3). The surfaces of PVC/Fe(DN)Mo, PVC/Cu(DN)Mo, and PVC/Zn(DN)Mo showed flat and dense structures, similar to that of PVC/(DN)Mo, while the cross-sections of PVC/Fe(DN)Mo, PVC/Cu(DN)Mo, and PVC/Zn(DN)Mo exhibited a typical tight and swollen structure, indicating the more stable structure of the char residue, which was consistent with the smoke density, MCC, and TGA results.

### 3.6. Mechanical Properties of PVC Composites

Figure 11 further assesses the influence of transition metal ion complexes on the mechanical properties of PVC. The impact strength, tensile strength, elongation at break, bending strength, and bending modulus of the PVC were 2.39 KJ/m^2^, 30.32 MPa, 68.31%, 41.25 MPa, and 1924.24 MPa, respectively. Compared with pure PVC, the impact strengths of PVC/(DN)Mo, PVC/Fe(DN)Mo, PVC/Cu(DN)Mo, and PVC/Zn(DN)Mo composites were slightly increased by 6%, 5%, 1%, and 4%, respectively. This was likely due to the fact that the transition metal complexes could passivate the front end of a crack, providing a slight toughening effect [16]. As shown in Figure 11b, a slight decrease in tensile strength and elongation at break appeared due to a good compatibility between transition metal complexes and the PVC, which prevented a remarkable decrease for adding so much fillers. As shown in Figure 11c, the bending strength and modulus of the four transition metal ion complexes showed satisfactory improvement. Combining the impact, tensile properties, and bending properties, it was demonstrated that the addition of the transition metal ion complexes could slightly strengthen and embrittle the PVC composites. On the other hand, the addition of transition metal ion complexes had little adverse effect on the mechanical properties of the PVC matrix due to good compatibility between the transition metal ion complexes and the PVC matrix.

To confirm compatibility between the transition metal ion complexes and the PVC matrix, the SEM micrographs of PVC, PVC/(DN)Mo, PVC/Fe(DN)Mo, PVC/Cu(DN)Mo, and PVC/Zn(DN)Mo were compared (Figure 12). It can be seen that the fracture of pure PVC was relatively flat and smooth, which was consistent with the low impact toughness. In contrast, the surfaces of PVC/(DN)Mo, PVC/Fe(DN)Mo, PVC/Cu(DN)Mo, and PVC/Zn(DN)Mo were rough and delaminated, which was in accordance with their slight improvement in impact strength. Dispersed pointlike particles, which were flame retardants and smoke suppressants, appeared on the surface of the PVC/(DN)Mo and PVC/Fe(DN)Mo (Figure 12b,c). This showed a slightly poorer compatibility between (DN)Mo or Fe(DN)Mo and the PVC matrix, compared with PVC/Cu(DN)Mo and PVC/Zn(DN)Mo. Since they were affected by compatibility, the impact strength and flexural strength of (DN)Mo and Fe(DN)Mo showed a slight fluctuation, as shown in Figure 11a,c. The surfaces of PVC/Cu(DN)Mo and PVC/Zn(DN)Mo were particle-free and consistent with the surface characteristics of the PVC, which indicated their good compatibility with the PVC matrix.

## 4. Conclusions

In this work, the four transition metal complexes of (DN)Mo, Fe(DN)Mo, Cu(DN)Mo, and Zn(DN)Mo were synthesized via photocatalytic copolymerization and an ion-exchange reaction. The results of XPS, XRD, and FTIR showed that the MoO_4_^2−^, Fe^3+^, Cu^2+^, and Zn^2+^ were ionically bonding with DN. The TGA and SEM results showed that these transition metal complexes possessed sufficient thermal stability and good dispersion. The MCC and LOI tests showed that Fe(DN)Mo, Cu(DN)Mo, and Zn(DN)Mo possessed better flame retardancy. The smoke density test showed the superior smoke suppression of Fe(DN)Mo, Cu(DN)Mo, and Zn(DN)Mo, and the TG test showed that these transition metal complexes could endow the PVC matrix with better stability. Cu(CN)Mo exhibited an overall better flame retardancy and smoke-suppression effect. It was also found that the addition of transition metal complexes promoted the formation of a dense carbon layer on the PVC surface during combustion, which could protect the interior PVC matrix. Owing to good compatibility of the four transition meatal complexes with the PVC matrix, they hardly decreased the mechanical properties of PVC.

## Figures and Tables

**Figure 1 materials-14-02634-f001:**
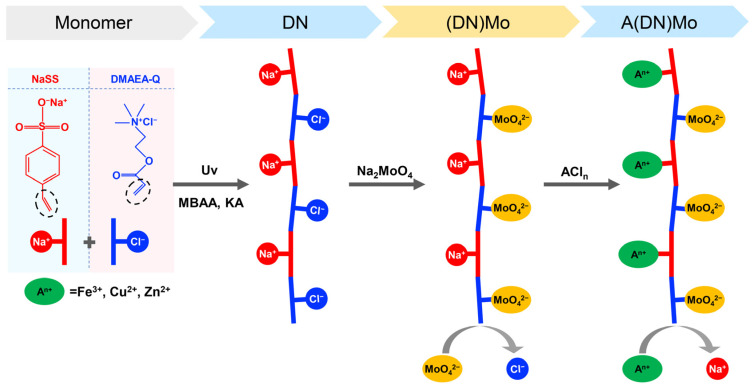
The synthetic route and molecular structure of transition metal ion complexes.

**Figure 2 materials-14-02634-f002:**
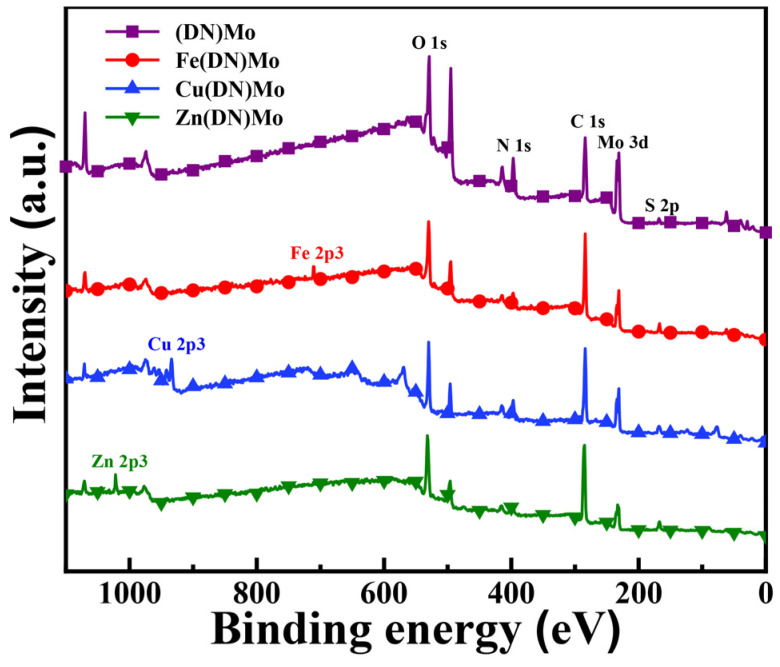
XPS spectra of (DN)Mo, Fe(DN)Mo, Cu(DN)Mo, and Zn(DN)Mo.

**Figure 3 materials-14-02634-f003:**
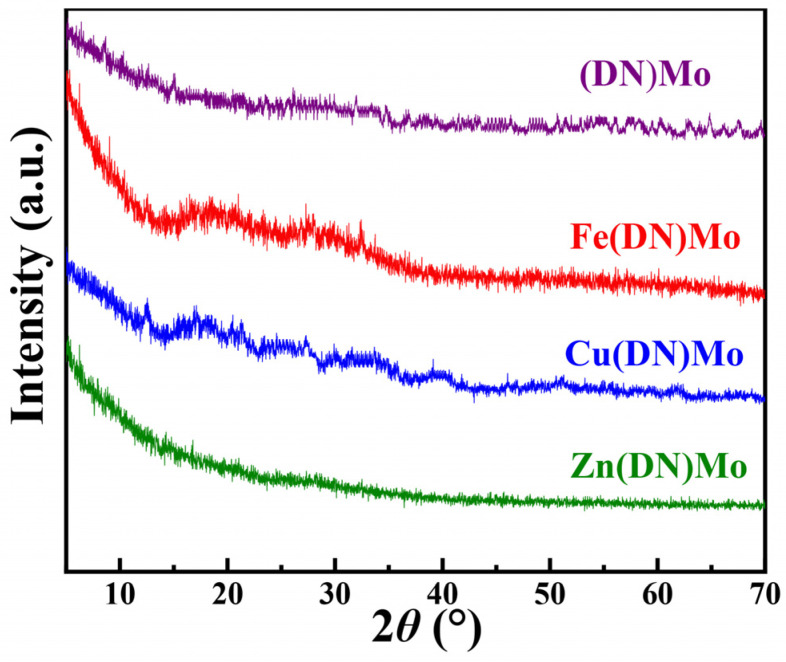
XRD diffraction patterns of (DN)Mo, Fe(DN)Mo, Cu(DN)Mo, and Zn(DN)Mo.

**Figure 4 materials-14-02634-f004:**
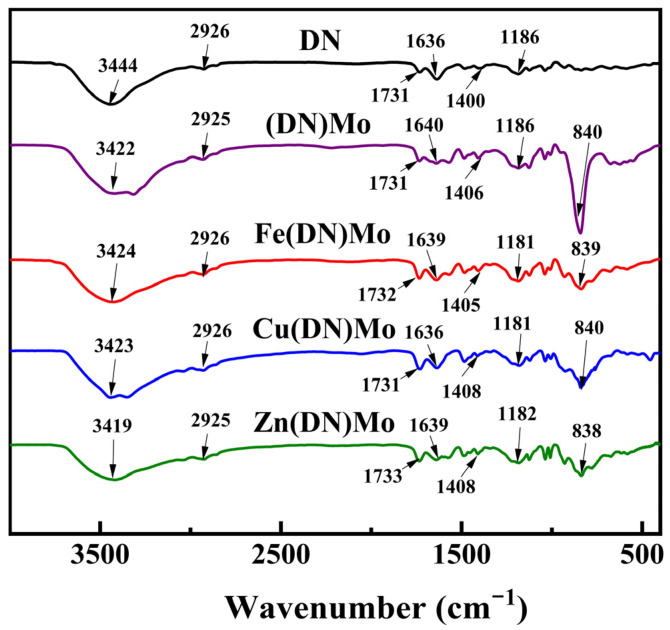
FTIR spectra of DN, (DN)Mo, Fe(DN)Mo, Cu(DN)Mo, and, Zn(DN)Mo.

**Figure 5 materials-14-02634-f005:**
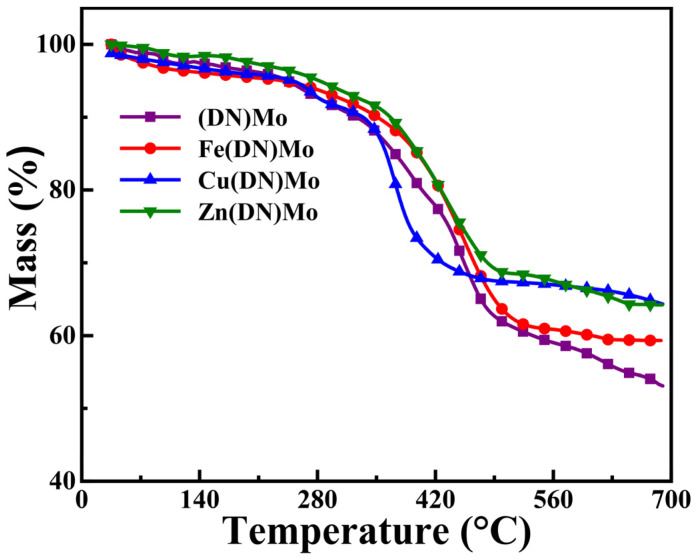
TG curves of (DN)Mo, Fe(DN)Mo, Cu(DN)Mo, and Zn(DN)Mo in N_2_.

**Figure 6 materials-14-02634-f006:**
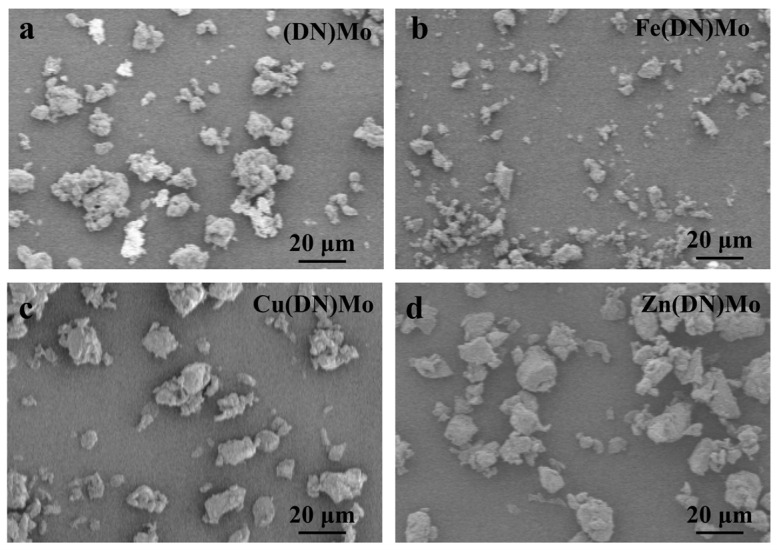
SEM micrographs of four transition metal ion complexes: (**a**) (DN)Mo, (**b**) Fe(DN)Mo, (**c**) Cu(DN)Mo, and (**d**) Zn(DN)Mo.

**Figure 7 materials-14-02634-f007:**
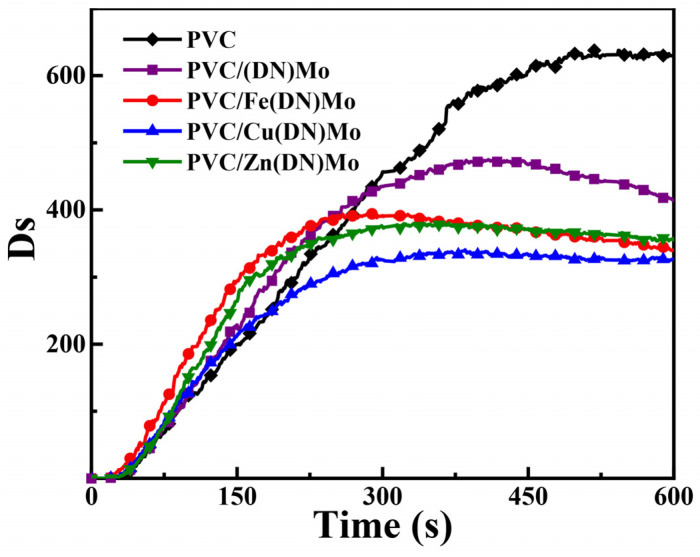
Function of Ds versus time for PVC, PVC/(DN)Mo, PVC/Fe(DN)Mo, PVC/Cu(DN)Mo, and PVC/Zn(DN)Mo.

**Figure 8 materials-14-02634-f008:**
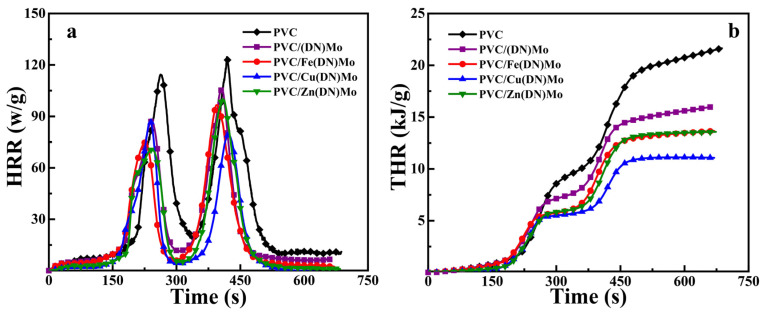
(**a**) HRR and (**b**) THR curves of PVC, PVC/(DN)Mo, PVC/Fe(DN)Mo, PVC/Cu(DN)Mo, and PVC/Zn(DN)Mo.

**Figure 9 materials-14-02634-f009:**
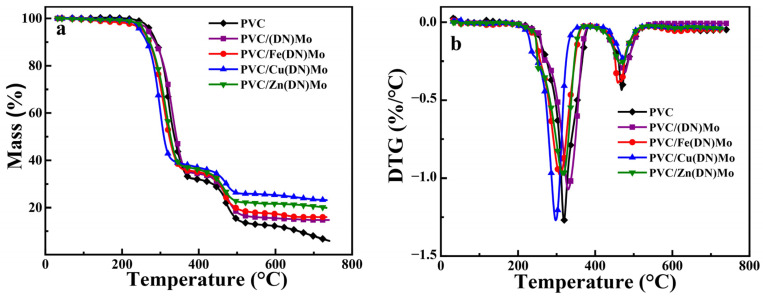
(**a**) TG and (**b**) DTG curves of PVC, PVC/(DN)Mo, PVC/Fe(DN)Mo, PVC/Cu(DN)Mo, and PVC/Zn(DN)Mo.

**Figure 10 materials-14-02634-f010:**
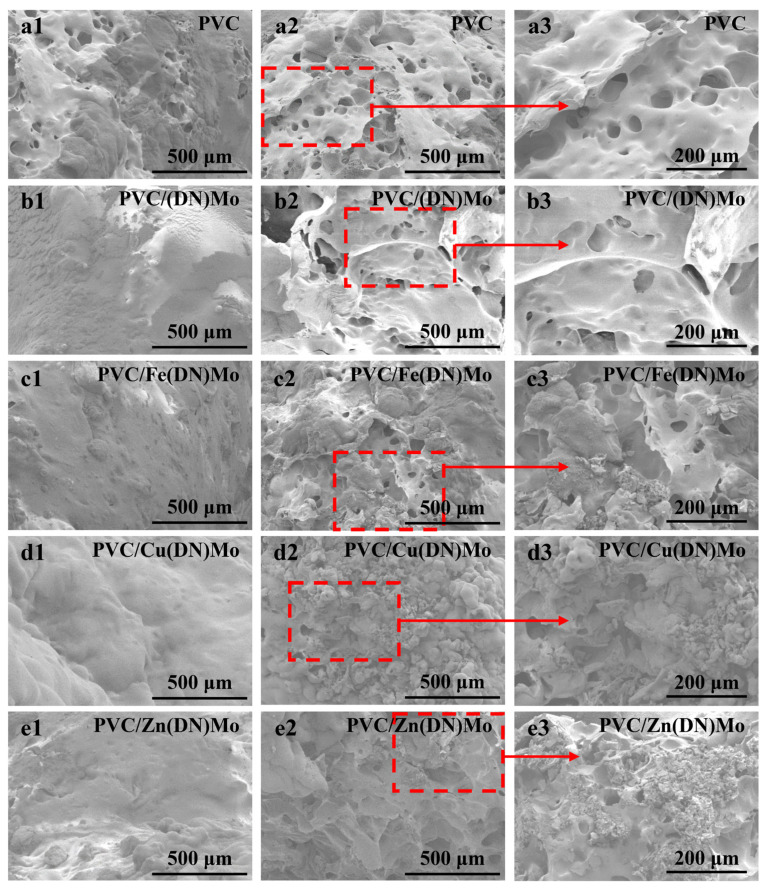
SEM micrographs of the residue chars of PVC, PVC/(DN)Mo, PVC/Fe(DN)Mo, PVC/Cu(DN)Mo, and PVC/Zn(DN)Mo. (**a1**), (**b1**), (**c1**), (**d1**), and (**e1**) are located at surface, while (**a2**), (**b2**), (**c2**), (**d2**), and (**e2**) are located in cross-sections. The local observation for (**a2**,), (**b2**), (**c2**), (**d2**), and (**e2**) are shown in (**a3**), (**b3**), (**c3**), (**d3**), and (**e3**), respectively.

**Figure 11 materials-14-02634-f011:**
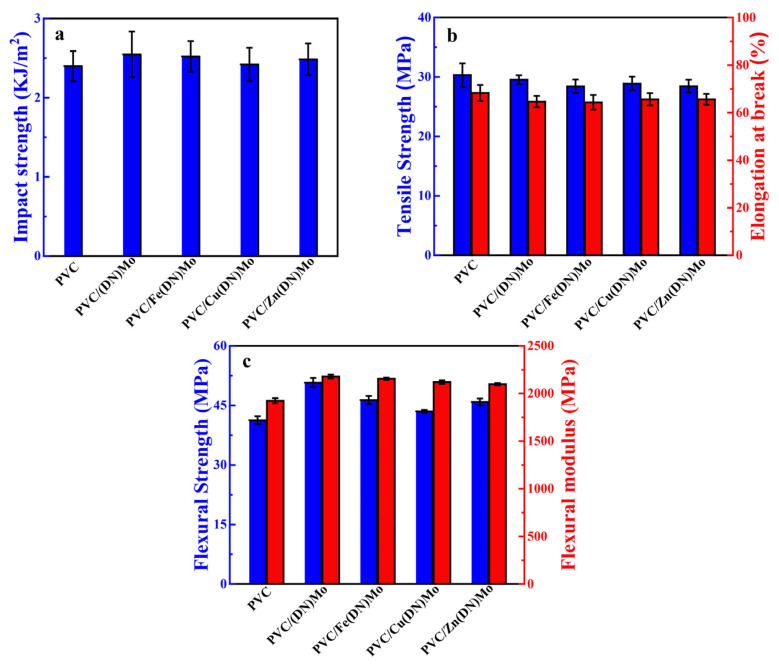
Mechanical properties of PVC, PVC/(DN)Mo, PVC/Fe(DN)Mo, PVC/Cu(DN)Mo, and PVC/Zn(DN)Mo: (**a**) impact strength, (**b**) tensile strength and elongation at break, and (**c**) flexural strength and flexural modulus.

**Figure 12 materials-14-02634-f012:**
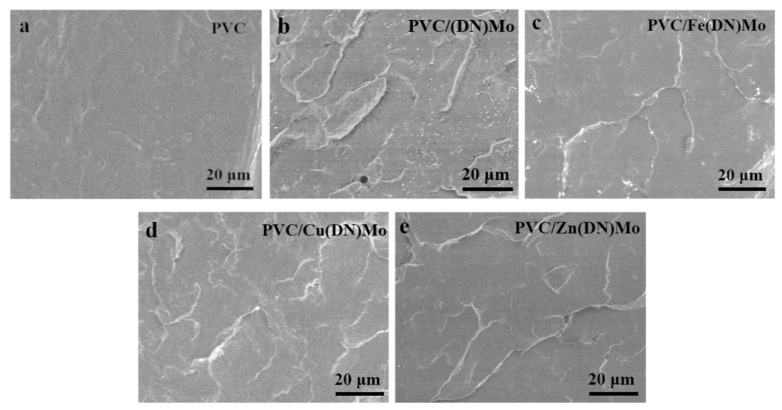
SEM micrographs of (**a**) PVC, (**b**) PVC/(DN)Mo, (**c**) PVC/Fe(DN)Mo, (**d**) PVC/Cu(DN)Mo, and (**e**) PVC/Zn(DN)Mo.

**Table 1 materials-14-02634-t001:** Ds_max_ and Ds_600_ of PVC, PVC/(DN)Mo, PVC/Fe(DN)Mo, PVC/Cu(DN)Mo, and PVC/Zn(DN)Mo.

Sample	Ds_max_	Ds_600_
PVC	637.7	633.4
PVC/(DN)Mo	474.8	404.9
PVC/Fe(DN)Mo	396.7	335.0
PVC/Cu(DN)Mo	340.3	324.7
PVC/Zn(DN)Mo	380.2	346.5

**Table 2 materials-14-02634-t002:** Combustion parameters of PVC, PVC/(DN)Mo, PVC/Fe(DN)Mo, PVC/Cu(DN)Mo, and PVC/Zn(DN)Mo.

Sample	First Stage		Second Stage		THR (kJ/g)	LOI (%)
	pHRR (w/g)	T_p_ (s)	pHRR (w/g)	T_p_ (s)		
PVC	114.43	262.5	124.64	418.5	21.66	35.2
PVC/(DN)Mo	87.19	240	106.08	407	15.97	37.2
PVC/Fe(DN)Mo	74.87	225.5	96.61	394	13.66	38.6
PVC/Cu(DN)Mo	86.93	237.5	79.21	422.5	11.07	39.1
PVC/Zn(DN)Mo	70.93	242.5	100.05	409.5	13.57	38.9

**Table 3 materials-14-02634-t003:** Thermogravimetric parameter value of PVC, PVC/(DN)Mo, PVC/Fe(DN)Mo, PVC/Cu(DN)Mo, and PVC/Zn(DN)Mo.

Sample	T_5%_ (°C)	First Stage		Second Stage		Char Residue (%)
		Mass (%)	T_max_ (°C)	Mass (%)	T_max_ (°C)	
PVC	273.71	66.05	319.02	27.12	468.66	5.94
PVC/(DN)Mo	270.12	62.16	332.81	23.01	480.06	14.78
PVC/Fe(DN)Mo	262.63	60.66	318.28	22.91	467.85	16.01
PVC/Cu(DN)Mo	254.92	58.27	305.75	18.43	478.63	23.14
PVC/Zn(DN)Mo	265.14	60.01	324.05	19.93	472.29	19.91

## Data Availability

All data, models, and code generated or used during the study appear in the submitted article.

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
