# Peer review of "Synthesis of Transition Metal Complexes and Their Effects on Combustion Properties of Semi-Rigid Polyvinyl Chloride"

_materials, 2021, doi:10.3390/ma14102634_

Round 1

Reviewer 1 Report

It is known that a fire produces large amounts of pollutants, often toxic, with a composition that is difficult to predict. For this reason, fire poses a threat not only to our property but also to health and even life. Under these conditions, more than half of people die in a fire not because of heat, but because of poisoning or suffocation. It is particularly dangerous under the conditions of PVC fire, which for safety reasons is excluded for use in many machines and structures. Therefore, the article is of great application importance.
Numerous flame-retardant substances for polymers are known, but most of them must be used in large excess, which adversely affects their mechanical properties.
The metal ion complexes produced in the work: (DN) Mo, Fe (DN) Mo, Cu (DN) Mo and Zn (DN) Mo used for the modification of PVC proved that, with the participation of 8 parts in the composite, they significantly improve the LOI without deteriorating mechanical properties. The authors achieved such an effect due to the correlation between the carbonization efficiency and the material's fire resistance. The porous layer of coke formed during pyrolysis acts as a barrier to the mass flow between the solid phase heated by the flame, but not burning, and the gas phase, which, while burning, provides the heat that heats the solid phase. For this reason, the coke layer lowers the surface temperature of the product, thereby reducing the rate of PVC pyrolysis and the associated rate of release of gaseous decomposition products, the combustible part of which feeds the flame. Coke separating the flame from the decomposing polymer thus blocks the inflow of free radicals formed in the gas phase during its combustion, and this changes the kinetics of chemical reactions. The flame is extinguished when the emission of flammable gases is less than necessary to sustain the fire. The Cu ions, which catalyzed the formation of coke earliest, were the most effective in this respect, as noted in the study. As has also rightly been pointed out, the barrier effectiveness of coke depends on its structure and physical properties. The most advantageous structure is a tight outer shell with a porous layer underneath which provides thermal insulation. Such layers have been produced in PVC / Cu (DN) Mo composites.
The work may be printed with minor linguistic corrections.
Table 3.5 should be renumbered to 3. References are cited correctly.

Author Response

Dear Editor.
Thank you very much for your suggestions. We have made further overall changes to the article. We have corrected the table numbering issue you raised, as shown in line 358 of the submission. Thank you very much for your professionalism and patience.

Reviewer 2 Report

The manuscript on the effect of transition metal complex insertion on the combustion properties of semi-rigid PVC shows some novelty in the synthesis part. The subject is of great interest in the research community and industry. However, the paper has some shortcomings and the interpretation of results needs some additions. In that sense, the Authors can see my recommendations to make their research better understood:

  • Line 36: Authors should add other references about the applications described here. Moreover, it is recommended that these new references be recent, since the one in the manuscript is from 2007.
  • Line 106: ... were synthesised? Authors should rewrite the synthesis procedure to make it easier to understand.
  • Line 125: How was the introduction of the transition metal ion complexes, X(DN)Mo done? What is the amount added? This Subtopic should be rewritten for ease of understanding.
  • Line 151: What were the conditions used to acquire the micrographs?
  • Line 193: It is not possible to see any peaks for (DN)Mo and Zn(DN)Mo as the Authors mention in the manuscript. Some peaks/bands are visible in difractogram of Fe(DN)Mo and Cu(DN)Mo, but the Authors should indicate what they correspond to.
  • Line 237: The curves should be presented in order to be possible to distiguish them, for instance, by changing the yy-axis, from 0 - 100 to 40 - 100 Mass (%).
  • Line 240: This conclusion is questionable. Since in each sample there are different sizes of floccules, with image magnification, it is not possible to see whether or not the larger floccules are from glomeration. Good dispersion is not seen by this technique. In SEM images it is possible to see the morphology of the dust/flakes/particles. It is recommended that the magnification of the images should be higher and to see the size of the flakes, the Authors try another technique, Dynamic Light Scattering for example.
  • Line 246: What does Ds mean?
  • Line 255 (Figure 7): Why was this test only done for 600s?
  • Line 263: Reference?
  • Line 377: It would perhaps be useful to see SEM images of PVC and all compounds before burning to compare the morphology in both situations. From Figure 10, it is only possible to understand the influence of the addition of the transition metal ion complexes. Are the images in Figure 12? If they are, the results should be discussed including those images, describing the preparation of the samples for image acquisition.

Obviously, after the suggested changes, I recommend that the Authors revise the Abstract and the Conclusion. 

Author Response

Dear Editor.
Thank you very much for your suggestion. We have made further overall changes to the article. In response to your suggestion we have made the following changes:

1 Q: Line 36: Authors should add other references about the applications described here. Moreover, it is recommended that these new Moreover, it is recommended that these new references be recent, since the one in the manuscript is from 2007.

A: We have added new references, and the corrected position is at line 36.

2 Q:Authors should rewrite the synthesis procedure to make it easier to understand.

A:We have re-described the experimental procedure in detail and illustrated it with pictures so that readers can better understand it (line 96).

3 Q:How was the introduction of the transition metal ion complexes, X(DN)Mo done? What is the amount added? This Subtopic should be rewritten for ease of understanding.

A:A 1 mol/L solution of Na2MoO4 was used in the experiment and as much MoO42- as possible was allowed to enter the DN (line 107).

4 Q:What were the conditions used to acquire the micrographs? (SEM)

A: The accelerating voltage in 5 kV, and the deposition time of sputtered-coated with a layer of gold is 20 s (line 152).

5 Q:It is not possible to see any peaks for (DN)Mo and Zn(DN)Mo as the Authors mention in the manuscript. Some peaks/bands are visible in difractogram of Fe(DN)Mo and Cu(DN)Mo, but the Authors should indicate what they correspond to.

A: In Figure 3, the noise in the XRD data is quite obvious, which indicates that the peak information is weak and corresponds to a minimal crystalline structure, which is consistent with our expectation. And the relevant literature is cited in line 208 in support.

6 Q: The curves should be presented in order to be possible to distiguish them, for instance, by changing the yy-axis, from 0 - 100 to 40 - 100 Mass (%)。

A:We have adjusted the range of the y-axis in Figure 5.

7  Q:This conclusion is questionable. Since in each sample there are different sizes of floccules, with image magnification, it is not possible to see whether or not the larger floccules are from glomeration. Good dispersion is not seen by this technique. In SEM images it is possible to see the morphology of the dust/flakes/particles. It is recommended that the magnification of the images should be higher and to see the size of the flakes, the Authors try another technique, Dynamic Light Scattering for example.

A:We obtained higher magnification SEM photographs and confirmed that no significant agglomeration occurred in Figure 6.

8  Q:What does Ds mean?

A:  Ds represents the smoke density. According to the ISO 5959-2: 2006  standard, Ds600 is the Ds value at 600s and Dsmax is the maximum DS value during the 600s of combustion (line 162). 

9 Q:Why was this test only done for 600 s?

A: According to the ISO 5959-2: 2006  standard,  this value for 600 s is of general reference.

10 Q: Reference?

A: The relevant references were added  (line 272).

11 Q: It would perhaps be useful to see SEM images of PVC and all compounds before burning to compare the morphology in both situations. From Figure 10, it is only possible to understand the influence of the addition of the transition metal ion complexes. Are the images in Figure 12? If they are, the results should be discussed including those images, describing the preparation of the samples for image acquisition.

A:  The synthesized samples, burned carbon layers, and PVC composites were obtained. Their sources and preparation methods are shown in line 152.

12 Q: The Authors revise the Abstract and the Conclusion. 

A: We have reworked the Abstract and Conclusions to better express our views.

Reviewer 3 Report

This manuscript focuses on the synthesis of organometallic compounds and on their use as flame retardant and smoke suppression agents in plasticized PVC. The authors claim their efficiency without mechanical properties detriment. The manuscript is well arranged but needs some corrections. Here are my suggestions.

  1. Please check the English in the whole manuscript, since many sentences (like that in lines 259-263) must be better formulated and many words corrected (see for instance lines 141-142, the uncorrected term “synthesized” in lines 106 and 133, as well as the use of “a amphoteric” in line 107 etc.) . Please, also ensure that abbreviations have been defined in parentheses the first time they appear (for instance the abbreviation phr is used in line 71 but its meaning is disclosed in line 125).
  2. In the section “ preparation of the PVC composites”, please specify the cooling conditions of the sheets.
  3. In Table 1 make clear the content of transition metal ion complexes, adjusting the units.
  4. In the sections 3.3 and 3.4 it is not mentioned the degradation of the transition metal ion complexes, although data show that it must start likely before the main PVC decomposition. The endothermic decomposition of transition metal ion complexes could also have a role in their flame and smoke retardant efficiency.

Author Response

Dear Editor.
Thank you very much for your suggestions and professionalism. We have made a thorough revision of the article.

1 Q:Please check the English in the whole manuscript, since many sentences (like that in lines 259-263) must be better formulated and many words corrected (see for instance lines 141-142, the uncorrected term “synthesized” in lines 106 and 133, as well as the use of “a amphoteric” in line 107 etc.) . Please, also ensure that abbreviations have been defined in parentheses the first time they appear (for instance the abbreviation phr is used in line 71 but its meaning is disclosed in line 125).

A: We have corrected the presentation and grammar of the entire article so that the reader can better understand our ideas.

2 Q: In the section “ preparation of the PVC composites”, please specify the cooling conditions of the sheets.

A: The cooling condition of sheets was added at line 125.

3 Q: Table 1 make clear the content of transition metal ion complexes, adjusting the units.

A:We have changed the structure of the table 1 and corrected the units.

4 Q: In the sections 3.3 and 3.4 it is not mentioned the degradation of the transition metal ion complexes, although data show that it must start likely before the main PVC decomposition. The endothermic decomposition of transition metal ion complexes could also have a role in their flame and smoke retardant efficiency.

A: The decomposition of transition ion complexes does contribute to their flame retardancy and smoke protection efficiency (line 353).

Round 2

Reviewer 2 Report

The Authors have made an impressive work after the suggestions submitted. I recommend this work to be accepted in the present form.

Reviewer 3 Report

Corrections have been made and the manuscript is now suitable for publication. I suggest to check again spelling, since I have noted that in many lines concetrate is written instead of concentration.